# Geochemical, Mineralogical and Morphological Characterisation of Road Dust and Associated Health Risks

**DOI:** 10.3390/ijerph17051563

**Published:** 2020-02-28

**Authors:** Carla Candeias, Estela Vicente, Mário Tomé, Fernando Rocha, Paula Ávila, Alves Célia

**Affiliations:** 1Geobiosciences, Geotechnologies and Geoengineering Research Centre (GeoBioTec), Department of Geosciences, University of Aveiro, 3810-193 Aveiro, Portugal; candeias@ua.pt (C.C.); tavares.rocha@ua.pt (F.R.); 2Centre for Environmental and Marine Studies (CESAM), Department of Environment, University of Aveiro, 3810-193 Aveiro, Portugal; estelaavicente@ua.pt; 3School of Technology and Management (ESTG), Polytechnic Institute of Viana do Castelo, Avenida do Atlântico, nº 644, 4900-348 Viana do Castelo, Portugal; mariotome@estg.ipvc.pt; 4LNEG, National Laboratory of Energy and Geology, Rua da Amieira, 4466-901 São Mamede de Infesta, Portugal; paula.avila@lneg.pt

**Keywords:** road dust, traffic, PM_10_ emission factors, enrichment index, human health risk

## Abstract

Road dust resuspension, especially the particulate matter fraction below 10 µm (PM_10_), is one of the main air quality management challenges in Europe. Road dust samples were collected from representative streets (suburban and urban) of the city of Viana do Castelo, Portugal. PM_10_ emission factors (mg veh^−1^ km^−1^) ranging from 49 (asphalt) to 330 (cobble stone) were estimated by means of the United Stated Environmental Protection Agency method. Two road dust fractions (<0.074 mm and from 0.0074 to 1 mm) were characterised for their geochemical, mineralogical and morphological properties. In urban streets, road dusts reveal the contribution from traffic emissions, with higher concentrations of, for example, Cu, Zn and Pb. In the suburban area, agriculture practices likely contributed to As concentrations of 180 mg kg^−1^ in the finest road dust fraction. Samples are primarily composed of quartz, but also of muscovite, albite, kaolinite, microcline, Fe-enstatite, graphite and amorphous content. Particle morphology clearly shows the link with natural and traffic related materials, with well-formed minerals and irregular aggregates. The hazard quotient suggests a probability to induce non-carcinogenic adverse health effects in children by ingestion of Zr. Arsenic in the suburban street represents a human health risk of 1.58 × 10^−4^.

## 1. Introduction

Particulates that are deposited on a road, usually called “road sediments”, “street dust” or “road dust”, are significant pollutants in the urban environment because they contain high levels of toxic metals and organic contaminants, such as polycyclic aromatic hydrocarbons [1,2]. These materials can be pulverised by the passing traffic and become aerolisable, making up a significant fraction of atmospheric particles. Another process contributing to the atmospheric particle loads is the resuspension of road dust, which is due to traffic induced turbulence, tyre friction or the action of the wind. Amato et al. [3] reported that local dust accounted for 7%–12% of the particulate matter <10 µm (PM_10_) concentrations at suburban and urban background sites in southern European cities and 19% at a traffic site, revealing the contribution from road dust resuspension. In the case of particulate matter <2.5 µm (PM_2.5_), the percentages decreased to 2%–7% at suburban and urban background sites and to 15% at the traffic site. Results for southern Spain indicated that road dust increased PM_10_ levels on average by 21%–35% at traffic sites, 29%–34% at urban background sites heavily affected by road traffic emissions, 17%–22% at urban-industrial sites and 9%–22% at rural sites [4]. In the urban area of Lanzhou, China, fugitive road dust was found to contribute to 24.6% of total PM_2.5_ emissions [5]. In Harbin, located at the north region of China, road dust represented 7% to 26% of PM_10_ [6]. In Delhi, road dust is the largest contributor to PM_10_, accounting for 35.6%–65.9% [7]. As motor exhaust emissions decay as a result of increasingly strict limits, the relative importance of emissions from resuspension and wear (brakes, tyres and road pavement) will grow. These emissions are recognised as non-exhaust sources. It has been estimated that in 2020 non-exhaust sources will represent about 90% of total road traffic emissions [8].

Long-term exposure to traffic-generated dust was estimated to cause every year 1.5 to 2 million premature deaths (mostly women and children) in low-income countries [9]. There is increasing awareness and concern of the potential adverse impacts of dust on health in high- and middle-income countries. A systematic literature review of articles on road dust and its effects on health was recently carried out by Khan and Strand [2]. The components of road dust particles have been associated with multiple health effects, in particular on the respiratory and cardiovascular systems. The list of health effects reported in the reviewed articles included chronic obstructive pulmonary disease, asthma, allergy, carcinoma, emergency cardiovascular disease issues, increased mortality due to cardiovascular disease, low birth weight and non-specific carcinoma. Numerous studies on the costs and efficacy of various products (e.g., suppressants) aimed at reducing road dust have been undertaken in the USA, Canada, Australia, New Zealand, South Africa and several European countries [9,10,11,12]. 

The amount of dust that is generated and then re-settles on the road surface depends on various factors including traffic speed, vehicle weight, local road conditions and rainfall. In regions with scarce precipitation, such as the Mediterranean countries, road dust resuspension is one of the major sources triggering PM_10_ exceedances. The composition of dust presents huge geographical and seasonal variations, so it is difficult to create a universal chemical fingerprint for this source [13]. To more accurately apportion the contribution of road dust to the atmospheric particulate matter levels, to better assess health risks, and to establish cause-effect relationships by seeking potentially toxic constituents, specific road dust chemical profiles should be obtained for each region or location. Moreover, to account for this source in pollutant inventories, appropriate emission factors are required. In Europe, some information on the chemical composition, mainly elemental, of resuspendable road dust samples from Barcelona, Girona and Zurich [14], four of the most polluted Polish cities [15], Oporto [1], and Paris [16], have been documented. This paper aims to present not only emission factors and the chemical composition, but also the mineralogical and morphological characteristics of road dust, as well as an assessment of health risks, in a medium-sized city with low-pollution levels for which no previous study has been conducted. 

## 2. Methodologies

### 2.1. Sampling

Road dust samples were collected in September and October 2018 on representative streets of Viana do Castelo (latitude: 41°41′35.63″ N; longitude: −8°49′58.33′ W), the most northern Atlantic city in Portugal with a population of about 40,000 in the most central urbanised area, although the municipality as a whole reaches 90,000 inhabitants. The city is located between Santa Luzia hill and the mouth of the river Lima. Its major industries are related to naval construction and repair, with one of the few large shipyards still in operation. It is also home to a large cluster of wind green electricity and car-parts industries. The old downtown streets are closed to traffic. 

Three streets were selected for road dust sampling (Appendix A): 

(S1) Suburban context—Rua Alto Xisto is a street with cobbled pavement made of granite cubes in a residential area on the outskirts of the city. One side of the street consists of terraced houses. The other side is flanked by an agricultural farm with some animals, and vineyard, corn and potato cultivation; 

(S2) Urban context—Local road within the campus of the Higher School of Technology and Management. It is composed of stone mastic asphalt pavement and located a few meters from the beach and the shipyards; 

(S3) Urban context—Avenida Combatentes da Grande Guerra is a central artery connecting to the train station. It is an avenue with shops, services and an elementary school. Its cobbled pavement is made of granite cubes. 

The selection of streets was carried out in collaboration with the city council. It was considered that the urban area could be subdivided into three sectors: (i) a residential area with a lower population density and some agricultural influence, (ii) a central area with a higher volume of traffic where the main commercial activities and public services take place, (iii) and another area on the coast line, also with busy streets, but with industrial influence. For each sector, a street representing the dominant pavements and traffic was chosen. The various samplings took place on 3 weekends in September and October 2018 and implied the traffic cut by the police authorities. Road dust was collected on delimited lane sections using a vacuum cleaner, following the procedure described by the United Stated Environmental Protection Agency (USEPA) in its AP-42 document [17]. In each street, several subsamples were obtained by vacuuming segments of 20 m in length and width corresponding to the lanes. The collection of the first subsample started at a distance of 50 m from the intersection with another street. Distances of approximately 50 m were maintained between sampled segments. The weight of subsamples ranged from about 200 to 600 g. For each road, a composite sample from a minimum of 3 incremental subsamples was created.

### 2.2. Geochemical, Mineralogical and Morphological Characterisation

After sampling, the vacuum cleaner bags were stored in polyethylene bags and brought to the laboratory, where the USEPA methodology for analysis of surface/bulk dust loading samples was followed [18]. Samples were wet sieved with addition of distilled water through standard Taylor screens, dried at ~40 °C, and weighed in a microbalance with 1 μg sensitivity. While particles >1 mm were rejected, the fraction <0.074 mm (F1) and that between 0.0074 and 1 mm (F2) were retained for further analyses.

The chemical composition of the road dust fractions was determined by X-ray fluorescence (XRF) using an Axios PW4400/40 X-ray (Marvel Panalytical, Almelo, The Netherlands) fluorescence wavelength dispersive spectrometer. The limit of detection (LOD) (i.e., the concentration at which it is possible to distinguish the signal from the background) is provided in Table 1. Mineralogy was determined by X-ray diffraction (XRD) using a X’Pert-Pro MPD diffractometer (Marvel Panalytical, Almelo, The Netherlands) with a Cu-Kα radiation at 45 kV, 40 mA, and with a step size of 0,02°2θ, in 2°–90° 2θ angle range. A Hitachi S-4100 scanning electron microscope (SEM) equipped with a Quantax 400 Energy Dispersive Spectrometer (EDS) (Bonsai Advanced, Madrid, Spain) was used for point and area chemical analyses. Particle size distributions of road dusts were determined by X-ray sedimentation technique with a Sedigraph III Plus grain size analyser (Micromeritics Instrument Corp., Norcross, GA, USA). This technique for determining the relative mass distribution of a sample by particle size is based on two physical principles: sedimentation theory (Stokes’ law) and the absorption of X-radiation (Beer-Lambert law). Precision and accuracy of analyses and digestion procedures were monitored using internal standards, certified reference material and quality control blanks.

### 2.3. Estimation of Emission Factors

Road dust emission factors (*EFs*) were estimated using the equation outlined in the AP-42A document of USEPA [19] for paved roads: *EF* = *k* (*sL*/2)^0.65^ × (*W*/3)^1.5^ − *C*,(1)
where:

*EF* = PM_10_ emission factor (g veh^−1^ km^−1^),

*k* = particle size multiplier for particle size range and units of interest (0.46 g veh^−1^ km^−1^ for PM_10_),

*s =* surface material silt content, defined as particles that pass through a 200-mesh screen, which corresponds to 74 µm (%),

*L =* surface material loading, defined as mass of particles per unit area of the travel surface (g m^−2^),

*W* = average weight (tons) of the vehicles travelling the road (a value of 2 tons was assumed),

*C* = emission factor for 1980’s vehicle fleet exhaust, brake wear and tyre wear (0.1317 g veh^−1^ km^−1^ for PM_10_).

### 2.4. Enrichment Index

The enrichment index (*EI*) of an element is based on its concentration and the concentration of a reference element. The latter is chosen based on its low occurrence variability in order to identify geogenic and anthropogenic contributions. Due to the abundant natural occurrence on Earth’s crust, Al was selected for this study. *EI* was calculated as follows:(2)$$EI_{x}=\left(X/C_{ref}\right)sample/\left(X/C_{ref}\right)crust,$$
with *EI_x_* is the enrichment index of the element *X*, *X* the concentration of the element and *C_ref_* the concentration of the reference element (Al). The Earth’s crust individual elemental composition was taken from Riemann and Caritat [20]. *EI* < 1 indicates no enrichment (natural contribution), 1 ≤ *EI* <3 minor anthropogenic enrichment, 3 ≤ *EI* < 5 moderate anthropogenic enrichment, 5 ≤ *EI* < 10 moderately severe anthropogenic enrichment, 10 ≤ *EI* < 25 severe anthropogenic enrichment, 25 ≤ *EI* < 50 very severe anthropogenic enrichment, and *EI* ≥ 50 extremely severe anthropogenic enrichment [21].

### 2.5. Human Health Risk Assessment of Exposure to Potential Toxic Elements in Road Dust

A human health risk assessment assumes that residents, both children and adults, are directly exposed to potential toxic elements through three different routes: ingestion, dermal absorption and inhalation of particles [22,23,24]. For road dust, it was assumed that intake rates and particle emission are similar to those established for soils. Equations (3)–(5) were used to estimate the chronic daily intake (*CDI*_route_, ingestion and dermal in mg kg^−1^·d^−1^; inhalation in mg m^−3^ for non-carcinogenic effects, and µg m^−3^ for carcinogenic effects) of each exposure route and for separated elements:(3)$$CDI_{ing-nc}\;=\frac{C\;\times IngR\times EF\times ED}{BW\;\times AT_{nc}\;}\times 10^{-6},$$
(4)$$CDI_{drm-nc}=\frac{C\times SA\times SAF\times DA\times EF\times ED}{BW\;\times AT_{nc}}\times 10^{-6},$$
(5)$$CDI_{inh-nc}=\frac{C\times InhR\times EF\times ED}{PEF\times BW\times AT_{nc}},$$
where, *C* (mg kg^−1^) is the concentration of the element in road dust, *IngR* is the ingestion rate (200 and 100 mg d^−1^ for children and adults, respectively), *InhR* is the inhalation rate (7.6 and 20 m^3^ d^−1^ for children and adults, respectively), *EF* is the exposure frequency (180 d yr^−1^), *ED* is the exposure duration (6 and 24 years for non-carcinogenic effects in children and adults, respectively, and 70 years is the lifetime for carcinogenic effects), *BW* is the average body weight (15 and 70 kg for children and adults, respectively), *AT_nc_* is the averaging time for non-carcinogenic effects *(AT* days = *ED ×* 365), *SA* is the exposed skin area (2800 and 5700 cm^2^ for children and adults, respectively), *SAF* is the skin adherence factor (0.2 and 0.07 mg cm^−2^ for children and adults, respectively), *DA* is the dermal absorption factor (0.03 for As and 0.001 for other elements), and *PEF* is the particulate emission factor (1.36 × 10^9^ m^3^ kg^−1^) [22,23,24,25,26].

For each element and route, the non-cancer toxic risk was estimated by computing the chronic hazard quotient (*HQ*, Equation (6)) for systemic toxicity [24]. A *HQ* less than or equal to 1 indicates that adverse effects are not likely to occur, and thus can be considered to have negligible hazard, while *HQ* > 1 suggests that there is a high probability of non-carcinogenic effects to occur. To estimate the overall developing hazard of non-carcinogenic effects, it is assumed that toxic risks have additive effects. Therefore, it is possible to calculate the cumulative non-carcinogenic hazard index (*HI*), which corresponds to the sum of *HQ* for each pathway (Equation (7)) [27].
(6)$$HQ_{route}=\frac{CDI_{route}}{R_{f}D_{route}},$$
(7)$$HI=\sum HQ=HQ_{ing}+HQ_{drm}+HQ_{inh},$$
with *R_f_D* being the chronic reference dose (values estimated as given in RAIS) [24].

The probability of an individual to develop any type of cancer over his lifetime (Risk), as a result of exposure to the carcinogenic hazards, was computed for each route according to Equation (8). The carcinogenic Total Risk is the sum of risk for each route (Equation (9)). A cancer risk below 1 × 10^−6^ is considered insignificant, being this value the carcinogenic target risk. A risk above 1 × 10^−4^ (a probability of 1 in 10,000 of an individual developing cancer) is considered unacceptable [22,23,24,27]:(8)$${Risk}_{route}=CDI_{route}\times CSF_{route},$$
(9)$$Total\;Risk\;=\;\sum Risk={\;Risk}_{ing}+{Risk}_{drm}+{Risk}_{ihn}=\;CDI_{ing-ca}\times CFS_{ing}+CDI_{inh-ca}\times IUR+\frac{CDI_{drm-ca}\times CSF_{ing}}{ABS_{gi}},$$
(10)$$CDI_{inh-ca}=\frac{c\times IngR_{adj}\times EF}{{AT}_{ca}}\times 10^{-6},$$
(11)$$IngR_{adj}=\frac{ED_{child}\times IngR_{child}}{BW_{child}}+\frac{\left(ED_{resident}-ED_{child}\right)\times IngR_{adult}}{BW_{adult}},$$
(12)$$CDI_{inh-ca}=\frac{C\times EF\times ET\times ED}{PEF\times 24\times AT_{ca}},$$
(13)$$CDI_{drm-ca}=\frac{c\times {DA}_{d}\times EF\times {DSF}_{adj}}{{AT}_{ca}}\times 10^{-6},$$
(14)$$DSF_{adj}=\frac{ED_{child}\times SA_{child}\times SAF_{child}}{BW_{child}}+\frac{\left(ED_{resident}-ED_{child}\right)\times SA_{adult}\times SAF_{adult}}{BW_{adult}}$$
where *CSF* is the chronic slope factor (ingestion, (mg kg^−1^ d^−1^)^−1^; dermal*, CSF_ing_/ABS_gi_*), *ABS_gi_* is the gastrointestinal absorption factor, *IUR* is the chronic inhalation unit risk ((µg m^−3^)^−1^), *DFS_adj_* is the soil dermal contact factor-age-adjusted (362 mg yr kg^−1^ d^−1^), *AT_ca_* is the averaging time carcinogenic effects (*AT* days = *LT* × 365) and *ET* is the exposure time (8 h d^−1^) [27].

## 3. Results and Discussion

### 3.1. PM_10_ Emission Factors

For granite cube paved streets, very close PM_10_ emission factors were found: 277 mg veh^−1^ km^−1^ (central avenue, S3) and 330 mg veh^−1^ km^−1^ (street on the outskirts of the city, S1). On the asphalt paved street (S2), a much lower PM_10_ emission factor was obtained (49 mg veh^−1^ km^−1^). Using an in-situ resuspension chamber, PM_10_ emission factors in the range 12.0–29.4 mg veh^−1^ km^−1^, averaging 18.6 mg veh^−1^ km^−1^, were determined for asphalt roads in the city of Oporto [1]. However, a much higher value of 1082 mg veh^−1^ km^−1^ was estimated for a cobbled pavement in the same city. According to the researchers, this great emission factor was due to the very high roughness of cobblestones, which may have promoted the build-up of road sediments. Moreover, the joints between granite cubes filled with soil may have constituted an additional source of resuspendable dust. Following the USEPA methodology, PM_10_ emission factors for typical urban roads in Milan within 13–32 mg veh^−1^ km^−1^ were recently reported, which were found to be in the central range of the literature values in Europe [28]. Based on micro-scale vertical profiles of the deposited mass, PM_10_ emission factors from 5.4 to 9.0 mg veh^−1^ km^−1^ were obtained at inner-roads of Paris, whilst a higher value was estimated for the ring road (17 mg^−1^ veh^−1^ km^−1^) [16]. Based on a literature review, Boulter et al. [29] summarised the available information on emissions from road dust resuspension for the United Kingdom, USA, central and northern Europe. Differences in emission factors may be associated with local factors, such as road pavement type, regularity of street washing and precipitation events. On the other hand, while in Southern Europe high solar radiation is usually recorded, favouring the mobility of deposited particles, winter sanding or salting of roads and the use of studded tyres in Scandinavian and Alpine regions may promote the enhancement of road dust loadings. Road PM_10_ emission factors documented for Europe are, in general, lower than those reported for the USA. However, it must not be forgotten that American databases are older than European ones. 

### 3.2. Geochemical Characterization of Dust and Enrichment Index

Road dusts are known for their heterogeneous mix with diverse natural and anthropogenic origins. The composition can vary depending on geographical location, resuspended soil, atmospheric deposition and anthropogenic sources, which include traffic related particles such as metallic components, eroded road pavement, but also building construction and demolition, and power generation [14,30].

Cluster analysis of the XRF results (Appendix A) of the studied samples (S1 suburban environment influenced by agricultural activities; S2 and S3 urban streets) confirmed the chemical difference between the two analysed fractions (F1 with samples <0.074 mm; F2 with sizes >0.074 mm and <1 mm). Elemental concentrations (Table 1) revealed Si as the most representative constituent, followed by Al > K > Fe > Ca > Na. The highest Si content was found in F2 (fraction >0.074 mm and <1 mm). The enrichment index (Appendix A) suggests a low influence of anthropogenic activities in both size fractions (*EI* < 1.5). Crustal erosion and parent materials (e.g., Variscan granite parent rock) may also influence the concentrations of other elements, such as Fe, Al, Mg, Mn, Rb, Na, Ti, Mo, V and Zr [31]. Previous studies suggest that heavy metals such as Mn, V, Cu, Fe, Ni, Pb and Zn are linked to traffic emissions [32,33]. Low enrichment indices (<3) were also obtained for manganese, indicating a major natural source. However, this element is used in fuel additives, aluminium based alloys offering a high corrosion resistance and formability, vehicle applications, such as inner panels, and heater and radiator tubes. 

Iron, with higher concentrations in fraction F1 (max = 39,895 mg kg^−1^), showed comparable results to those obtained for road dusts in Xi’an [34], Shanghai [35], Hong Kong [35], Beijing [35], Delhi [36] and Madrid [37], suggesting a significant geogenic contribution, which is confirmed by *EI* < 3. Elements such as Al, Fe, Ti, Zr and Na have a potential source in soil resuspended dusts and marine spray (*EI* < 3). The anthropogenic contribution may be linked to the application of these elements in the production of metal alloys commonly used in vehicles (e.g., Fe as a major component of steel and associated rust); Al alloys associated with other elements such as Mg, Mn and Cu to reduce vehicle weight; alloys with Mn to avoid corrosion and deformation; sulphides (such as Sb_2_S_3_, MoS_2_ and SnS) and sulphates such as barite (BaSO_4_) applied in brakes; titanium oxide (TiO_2_) used as a pigment in white paints; and aluminium oxide and iron oxides also employed in brakes [33,38]. The detection of Zr may be, in part, related to vehicle exhaust emissions, since ceria/zirconia (CeO_2_/ZrO_2_) mixed oxides have become an essential component of three-way catalysts [39]. Asphalt materials are usually rich in Al, Si, K, and Ca, with smaller amounts of Fe, S, Mg, Zn, and Ti [40]. Elements such as P and K are commonly used in agriculture activities (e.g., phosphate, potassium nitrate). Higher concentrations of these elements were found in samples from the suburban location, indicating the contribution of the surrounding agricultural environment to road dust. The median concentrations of lead, chromium and copper in samples of urban streets, particularly in F1 fraction, were similar to those of other studies (e.g., Thessaloniki [41], Shiraz [42], Urumqi and Zhuzhou [43]). Lead, chromium and copper concentrations in the ranges 48–375, 2.0–498 and 47–995 mg kg^−1^, respectively, have been reported for street dusts of different cities on various continents [44], and references therein.

The enrichment index of Zn in the finest fraction (F1) ranged from 10.6 to 43.5, with a moderate (suburban location) to very severe anthropogenic enrichment (urban streets). Zn has been reported to be present as a minor constituent of silicate minerals and linked to fly-ash particles and to traffic related materials. Tyre rubber (ZnO and Cu/Zn layers formed during vulcanisation) and break wear resuspended particles are major sources of Zn, together with Cu, in urban areas [45]. Zn is also used as engine oil additive. Sternbeck et al. [46] suggested that fuel combustion may be a significant source of Zn. Nickel is a ubiquitous natural metal, used in the production of stainless steel and other nickel alloys with high corrosion and temperature resistance. It is considered a tracer of oil combustion [47]. The high concentration of Ba is likely related to brake wear since BaSO_4_ (barite) is used as filler for brake lining materials [48].

Antinomy concentrations revealed a very high enrichment index (F1 = 29.3; F2 = 158) in road dust from the central avenue (S3). This is a busy street with frequent braking, especially in front of the elementary school. Since there is no dedicated parking, parents stop at the lane to drop off or pick up their children. Antimony increases the hardness and mechanical strength of lead and is a significant metallic component in brake linings. It is also used in batteries and antifriction alloys, as additive in some types of oils, and applied in semiconductors and Sb_2_O_3_ in rubber vulcanisation flame retardants [40,49]. Cooper in the coarser fraction (F2) of samples from urban streets showed an *EI* of 73.6 and 18.6, while the corresponding value for the suburban location was 5.3. This element is commonly associated with traffic related activities, e.g., a component of brake pads wear. A mean Cu/Sb concentration ratio of 6.3 was obtained, in line with the 4.6 ± 2.3 diagnostic criteria proposed by Sternbeck et al. [46] for brake wear particles.

The maximum lead concentration in the coarse fraction did not exceed the minimum value recorded in the finest fraction. The highest concentration was found in the city centre (310 mg kg^−1^), while the minimum was recorded in the suburban area with rural influence (81 mg kg^−1^). According to Ferreira [50], the concentration of Pb in natural soils in this region is in the range 30–45 mg kg^−1^. Thus, the higher values observed in the present study suggest anthropogenic sources. This element presented an *EI* of 22.1 in fraction F2 from urban locations, whilst a value of 4 was determined for the suburban area. Lead and lead compounds are used in batteries and as pigments in paints. It has been documented that lead weights, which are used to balance motor vehicle wheels, are lost and deposited in urban streets, that they accumulate along the outer curb, and that they are rapidly abraded and ground into tiny pieces by vehicle traffic [51]. Lead oxide is a component of brake friction materials [15]. Its elevated concentrations in urban dust could be a consequence of common use of PbO_4_ as a gasoline additive in Portugal until the year 2000. Most of Pb is bound to stable fractions and only a negligible percentage is mobile, which contributes to its long persistence in the environment [15,45]. Tin revealed high *EI*, ranging from 11.6 in road dust samples from the suburban location to 36.3 in samples from urban streets. Tin was found to be one of the major elements in bulk brake pad samples, as well in airborne nano/micro-sized wear particles released from low-metallic automotive brakes [52]. Urban samples revealed a high W anthropogenic enrichment (*EI* up to 20.9). Tungsten is linked to break pads and tyre wear [53]. The abundant use of Br as flame retardant in several types of materials may explain the high *EI* of this element (23.3 and 47 in F1 of urban samples) [54].

An arsenic concentration of 180 mg kg^−1^ (*EI* = 72.7), six times higher than in urban samples, was observed in fraction F1 of the suburban sample. Arsenic rich particles in this road dust sample may originate from resuspended agricultural soil. It should be borne in mind that the street where the sampling took place is flanked by a farm. In addition to the abundant natural origin, there are many agricultural sources of arsenic to the soil, from pesticide application, livestock dips, organic manure to phosphate fertilisers [55]. Elemental As is used in the manufacture of alloys, particularly with lead (e.g., in lead acid batteries) and copper. Arsenic compounds are also extensively used in the semiconductor and electronics industries, in catalysts, pyrotechnics, antifouling agents in paints, among others. Fossil fuel combustion is considered a major contributor of anthropogenic As emissions to the air (mainly As^III^). Arsenic is present in the air of suburban and urban areas mainly in the inorganic As^V^ form. Background concentrations in soil range from 1 to 40 mg kg^−1^, with a mean value of 5 mg kg^−1^ [56], much lower than the levels found in this study.

### 3.3. Mineralogical Composition

The mineralogy of the two particle size fractions of the road dust samples was compared. X-ray diffraction results showed that road dust samples consist primarily of mineral matter accounting for a minimum of ~60%. The most abundant mineral was quartz [SiO_2_], especially in the coarse fraction. Quartz has higher structural hardness than other minerals preventing physical weathering and abrasion of road surfaces. Other minerals also present were muscovite [KAl_3_Si_3_O_10_(OH)_2_], albite [NaAlSi_3_O_8_], kaolinite [Al_2_Si_2_O_5_(OH)_4_], microcline [KAlSi_3_O_8_], Fe-enstatite [(Fe,Mg)_2_Si_2_O_6_] and graphite [C]. Minor proportions of calcite [CaCO_3_] and rutile [TiO_2_] were also observed, mostly in the coarse fraction of road dust collected in the suburban location. Clay forming minerals increase with the decrease of particle size. A significant proportion of amorphous content was detected in all samples, particularly in the finest fraction. This content may originate from weathered minerals, clays with low detection limit, organic matter and anthropogenic sources [57].

The bulk composition suggests that the SiO_2_ content of all samples reflects the abundance of leucocratic phases (quartz and feldspar) in the coarser fractions of road dust, while Fe, Ti and Mg indicate the higher abundance of melanocratic phases (biotite, garnet and hornblende) in the finer fractions. The ternary diagram Al_2_O_3_/CaO+Na_2_O+K_2_O/FeO+MgO (Figure 1) [58] of the studied samples defines three compositional trends, suggesting slightly different sources for each location. The composition of road dust from the suburban street (S1) reveals a feldspars/muscovite trend, S2 (urban) plots in the feldspars/biotite, and S3 (central avenue) in the feldspars/biotite with more compositional Ca. Feldspars are the most abundant mineral in all fractions. The ternary diagram Al_2_O_3_/CaO+Na_2_O/K_2_O suggests that all samples are closer to K-feldspar composition than plagioclase, and present also high content of micas (lever rule). This graphical analysis is in line with the XRD analysis, being quartz the most abundant mineral that is not plotted in these ternary diagrams.

The chemical index of alteration (CIA = (Al_2_O_3_/(Al_2_O_3_+CaO*+Na_2_O+K_2_O)) × 100, where CaO* is CaO if CaO < Na_2_O, but if CaO > Na_2_O, CaO = Na_2_O) [58,59] indicates: (a) low weathering if 50–60; (b) intermediate weathering if 60–80; and (c) intense weathering when >80 [60]. The coarser fraction of road dust from urban streets (S2 and S3) reveal a low weathering, with CIA = 57.3 and 58.4, respectively. The same road dust fraction of the sample collected at the suburban location (F2 of S1) indicate a low intermediate weathering with CIA = 61.6. The finest fractions (F1) of all samples present intermediate weathering, ranging from 61.3 to 67.6.

To assess the alumina abundance compared to other major cations, the index of compositional variation (ICV = ((Fe_2_O_3_+K_2_O+Na_2_O+CaO+MgO+MnO+TiO_2_)/Al_2_O_3_) was calculated [61,62]. Minerals such as kaolinite, illite and muscovite present ICV < 1, whereas minerals such as plagioclase, K-feldspar, biotite, amphiboles and pyroxenes have ICV > 1. The road dust sample from the suburban location influenced by agricultural activities, for both size fractions, presented an ICV < 1 (0.84 and 0.92), while the urban samples S2 and S3 revealed an ICV > 1 for both fractions (1.06 to 1.73), indicating an enrichment in rock forming minerals. The limitation of using the elemental composition to identify road dust sources should be noted, as not only the natural but also the anthropogenic contribution must be considered.

### 3.4. Morphology

The SEM analysis revealed the presence of well-formed minerals and irregular aggregates, with abundant silicate minerals and an un-sorted mixture of geogenic and anthropogenic particles (Figure 2). Particles with irregular and subangular to angular shapes, including plate like morphology, with variable chemical composition that comprises Fe, Cu, Zn, S, Al, Ti and Sb, suggest abrasion processes on their formation, such as tyre, break and pavement wear and vehicle corrosion interfaces. Rounded, longish and plate-like particles were also found. Although some of the particles present a larger size, they are formed by several smaller particles, usually <10 µm, which are composed of a mixture of anthropogenic and natural substances. It is known that brake abrasion generates particles containing Zn, Cu, Ti, Fe, Cu, and Pb, and other specific compounds such as sulphate silicate and barium sulphate. Particles from tyre abrasion comprise Cd, Cu, Pb, and Zn [63,64]. Silicates and Fe oxides/hydroxides tied to Cl/S can also be associated with traffic or resuspension. Particles with spherical morphology are produced in high-temperature processes (e.g., asphalt and industry/metallurgy). Silicate and iron plerospheres and cenospheres in fly ashes were also found, with typical Si-Al-Fe-Cu-Ca and Fe-Si-Al-Na compositions. Although carbon could not be calculated due to the SEM analysis technique, its presence was confirmed by the XRD analyses. Asphalt paving, or bituminous, materials are mainly made of carbonaceous components (e.g., saturated hydrocarbons, aromatics, asphaltenes, etc.) that are mixed with mineral aggregates. In addition to metals, particles from brake-pad and brake-disc abrasion consists of carbon fibres and graphite, while particulate material from tyre wear comprises various carbonaceous constituents (organic compounds such as natural rubber copolymer, organotin compounds, and soot) [64]. Particle size indicates that road dust might be an important source of resuspended atmospheric particulate matter associated with non-exhaust emissions (e.g., brake, tyre, and road wear), as suggested in previous studies [65]. The size of most of the non-exhaust particles is commonly much larger than that of exhaust particles, since its formation comprises processes such as corrosion, crushing and mechanical abrasion.

### 3.5. Grain Size Distribution

The grain size distribution of road dusts is presented in Figure 3. Results show a similar pattern for both suburban (S1) and urban areas (S2 and S3) with a marked unimodal distribution. The mass volume of particles peaked in the range from 10 to 106 µm, although small modes, barely noticeable, were observed below 5 µm. Particles < 100 µm can easily be resuspended in the wake of passing traffic or by the blowing wind and might enter the mouth and nose while breathing. Particulate matter of 10 and 2.5 µm or less in diameter (PM_10_ and PM_2.5_, respectively) can get deep into the lungs and some may even get into the bloodstream. Urban sample S3 presented a higher percentage of PM_10_, with S3_PM10_ 37.8% > S1_PM10_ 27.2% > S2_PM10_ 25.0%, while PM_2.5_ represented a smaller fraction, with 4.5% (S3) > 3.0% (S1) > 2.7% (S2) of the total mass volume. A literature review by Grigoratos and Martini [66] documented unimodal mass size distributions of brake wear PM_10_, with a mass weighed mean diameter of 2–6 μm. On the other hand, tyre wear PM_10_ often exhibits a bimodal distribution with one peak lying within the fine particle size range and the other one within the coarse range (5–9 μm). It is estimated that almost 40%–50% by mass of generated brake wear particles and 0.1%–10% by mass of tyre wear particles is emitted as PM_10_. In terms of mass, more than 85% of diesel particulate exhaust emissions are below 1 µm. Gasoline vehicles emit an even higher proportion of smaller particles than diesel vehicles [67]. 

### 3.6. Human Health Exposure Assessment

The hazard quotient (*HQ*), or non-carcinogenic effects, suggest that ingestion is the major route of children’s exposure to road dusts, with *HQ_ing_* ≈ *HI* (Figure 4), both by hand-to-mouth common habits and by resuspended particles. Dermal and inhalation routes can be considered negligible. Fraction F1 revealed a higher probability to induce non-carcinogenic health effects in children. The *HI* values for adults were approximately an order of magnitude lower than those for children. For both children and adults, Zr is the element that most contributes to possible non-carcinogenic effects. Adverse health effects may occur mostly by ingestion of resuspended particles (children: F1 *HQ_i_*_ng-Zr_ ≈ *HI* ranging from 27.51 to 553.10; F2 *HQ*_ing-Zr_ ≈ HI ranging from 5.66 to 8.71; adults: F1 *HQ*_ing-Zr_ ≈ HI ranging from 2.52 to 5.13; F2 *HQ*_ing-Zr_ ≈ HI ranging from 0.38 to 0.69). Studies suggest that Zr and its compounds represent a risk for pulmonary health effects (benign) potentially associated with short-term exposure [68]. 

The probability of an individual to develop any type of cancer over lifetime (Risk) by As (Risk_As_) content in fraction F1 of the suburban sample is 1.58 × 10^−4^ (Table 2), a Risk above the acceptable target of 1 × 10^−4^ proposed by USEPA [24], so the adoption of local measures is suggested. Fraction F2 of the same sample showed a Risk_As_ = 1.58 × 10^−5^. Risk_As_ values of 3.07 × 10^−5^ and 2.46 × 10^−5^ were obtained for fraction F1 of road dust from urban streets, while the corresponding values for fraction F2 were 1.84 × 10^−5^ and 1.05 × 10^−5^. These cancer risks are in the range for which management measures are required. The dermal risk for As in all samples and fractions was also within this range. Arsenic and its inorganic compounds are classified as carcinogenic to humans since 2012 [69]. The Pb content in fraction F1 of sample S3 is also indicative that, by the ingestion pathway, it may pose a risk (2.24 × 10^−5^) to human health. 

## 4. Conclusions

Road dust resuspension is one of the major sources causing PM_10_ exceedances, with detrimental effects on climate and human health. It has been demonstrated that emission inventories must use locally determined emission factors as these vary not only with weather and road conditions, but also with vehicle fleet. This study offers the first experimental estimates for road dust emission factors in the region of Viana do Castelo. A PM_10_ emission factor of 49 mg veh^−1^ km^−1^ was derived for an asphalt-paved road, which is in the range of values documented for the same type of pavement in a few other southern European cities, while much higher emissions (around 300 mg veh^−1^ km^−1^) were found for cobble stone streets. Although sampling took place in only three streets, these were carefully selected to represent different sectors of the city. On the other hand, because composite samples were obtained from multiple road segments and the fact that the emission factors of the present study are similar to those determined in a previous work for several streets in Oporto, also an Atlantic city with comparable pavements and traffic fleet, the representativeness of the values now estimated is broadened.

The chemical composition and the enrichment indices suggest an anthropogenic contribution of traffic-related elements, such as Br, Cl, Cr, Cu, P, Pb, S, Sn, W and Zn, especially to the finest road dust fraction (F1, <0.074 mm) from urban streets. In these locations, the fingerprint of marine spray is also identified with higher concentrations of Cl, Na and Mg. Samples from the suburban area presented an extremely severe anthropogenic enrichment in the finest fraction for the element As, possibly linked to agricultural activities and fossil fuel combustion. Quartz was mostly present in the coarser fraction, in which it was the most abundant mineral. Other minerals derived from natural sources were also observed (muscovite, albite, kaolinite, Fe-enstatite and graphite), as well as a significant amount of anthropogenic-related materials (amorphous). SEM and EDS analyses indicated that the main constituents in the amorphous content originated from non-exhaust traffic sources (brake, tyre and road abrasion) and from fuel combustion.

The estimation of non-carcinogenic health risks due to exposure to heavy metals indicated that children may experience adverse effects due to ingestion of the finest size fraction of road dust. For the suburban location, the risk associated with the ingestion of this finest road dust was above the acceptable target proposed by USEPA. Other samples and size fractions presented a risk by ingestion and dermal contact within a range indicating that management measures are required. 

## Figures and Tables

**Figure 1 ijerph-17-01563-f001:**
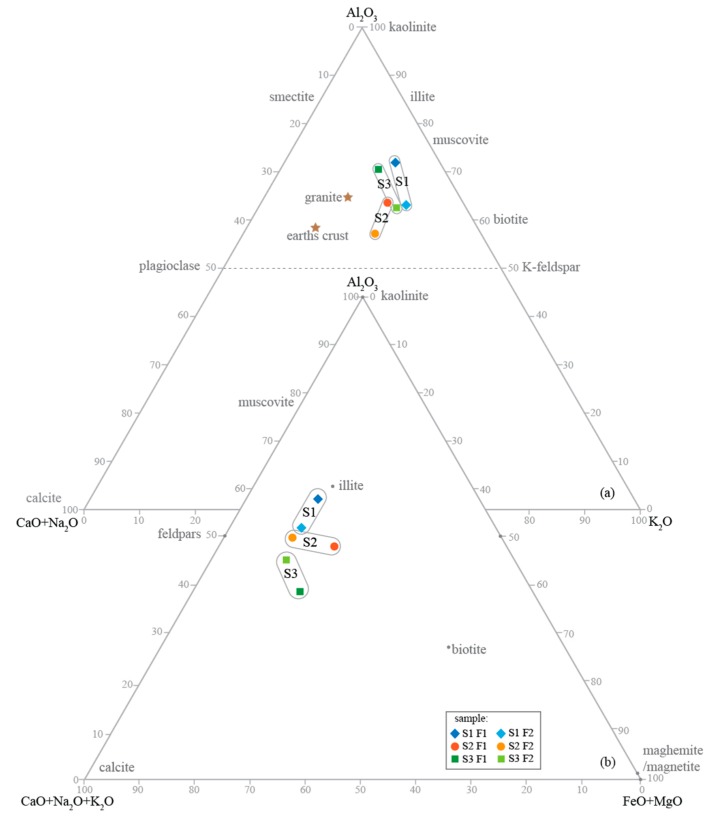
Ternary diagrams (a) Al_2_O_3_—CaO+Na_2_O+K_2_O—FeO+MgO, and (b) Al_2_O_3_—CaO+Na_2_O+K_2_O—FeO+MgO of the studied samples (adapted from [58]). Stars plotted indicate the average composition of the Earth’s crust and granite [20]. S1—suburban street, cobbled pavement; S2—urban street, asphalt pavement; S3—central avenue, cobbled pavement.

**Figure 2 ijerph-17-01563-f002:**
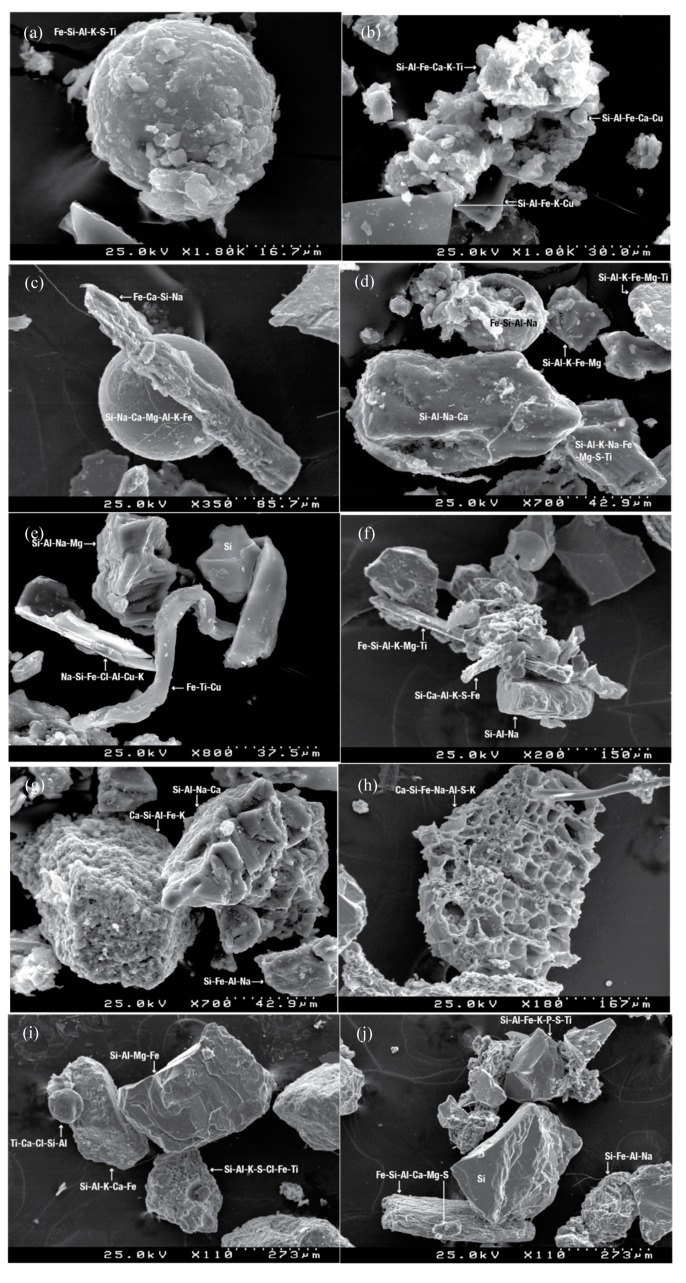
SEM images and composition of anthropogenic and geogenic road dust particles: (**a**–**f**) plerospheres, cenospheres, aggregates, plate like particles and fibrous steel with considerable plastic deformation and fragmentation; (**g**) irregular aggregates; (**h**) porous Ca-Si-Fe rich particle; (**i**,**j**) well-formed minerals.

**Figure 3 ijerph-17-01563-f003:**
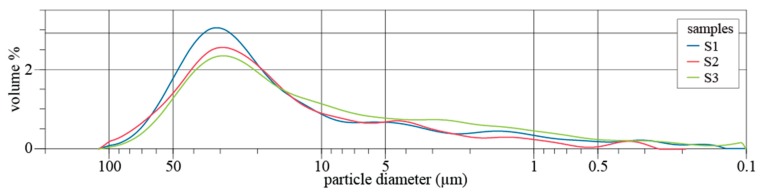
Size distribution (%) of road dust particles, Ø < 106 µm.

**Figure 4 ijerph-17-01563-f004:**
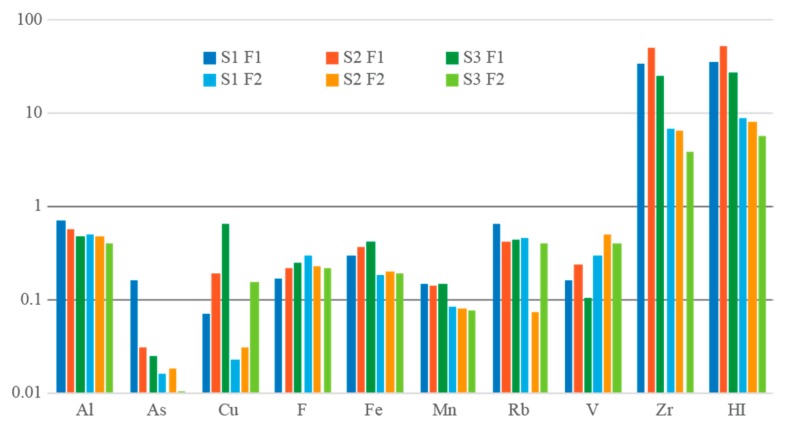
Non-carcinogenic chronic hazard quotient (*HQ*) of Al, As, Cu, F, Fe, Mn, Rb, V and Zr by ingestion in children and cumulative non-carcinogenic hazard index (*HI*). Logarithmic scale.

**Table 1 ijerph-17-01563-t001:** Summary descriptive statistics of elements analysed in two size fractions of road dusts from three selected locations. Concentrations in mg kg^−1^.

	LOD	F1		F2	
		min–max	Med ± SD	min–max	Med ± SD
**Al**	0.50	64,013–95,874	75,836 ± 13,150	54,873–67,749	66,241 ± 5748
**As**	4.06	28–180	35 ± 70	12–21	18 ± 3.5
**Ba**	6.90	270–560	390 ± 119	250–490	290 ± 105
**Br**	0.78	37–62	49 ± 11	7.7–17	11 ± 4.0
**Ca**	0.50	17,146–58,412	19,669 ± 18,887	9,806–27,938	13,687 ± 7795
**Cl**	0.50	3090–11,610	6890 ± 3485	220–400	330 ± 74
**Cr**	1.96	48–230	210 ± 81	49–67	60 ± 7.5
**Cu**	2.84	95–870	260 ± 333	31–210	42 ± 82
**F**	0.60	920–1350	1170 ± 176	1160–1640	1220 ± 214
**Fe**	0.50	27,613–39,895	35,188 ± 5059	17,353–18,633	18,416 ± 559
**Ga**	0.94	14–18	15 ± 1.5	9.6–12	10 ± 0.8
**K**	0.60	31,969–48,614	43,318 ± 6943	38,146–47,095	43,102 ± 3661
**Mg**	0.50	5947–6887	6610 ± 395	2883–5030	3920 ± 877
**Mn**	5.62	472–488	480 ± 6.3	248–279	263 ± 13
**Mo**	0.78	1.3–2.9	2.1 ± 0.7	2.9–11	6.9 ± 3.3
**Na**	0.50	10,416–14,214	10,638 ± 1740	8583–16,454	10,349±3372
**Ni**	2.00	15–29	16 ± 6.2	4.4–8.7	5.7 ± 1.8
**P**	0.60	3561–5158	3 749 ± 713	1554–2453	1667 ± 400
**Pb**	1.72	81–310	86 ± 107	37–79	40 ± 19
**Rb**	0.64	230–350	240 ± 54	200–250	220 ± 21
**S**	0.50	2908–19,384	4870 ± 7348	1173–1970	1950 ± 371
**Sb**	4.18	nd	nd	5.1–35	5.2 ± 14
**Si**	0.50	175,454–239,759	229,176 ± 28,153	312,012–320,823	316,480 ± 3597
**Sn**	3.02	35–75	36 ± 19	19–25	23 ± 2.6
**Sr**	0.72	90–250	190 ± 66	55–350	76 ± 134
**Ti**	0.50	3717–5809	4310 ± 880	2668–2836	2668 ± 79
**U**	1.22	3.5–4.1	3.6 ± 0.3	8.5–13.2	11.2 ± 1.9
**V**	2.78	10–23	15 ± 5.2	28–47	38 ± 7.6
**W**	3.70	nd	nd	7.7–25	16 ± 8.7
**Zn**	1.28	680–1870	1180 ± 488	160–510	220 ± 153
**Zr**	0.80	270–550	360 ± 117	41–74	70 ± 15.0

F1—fraction <0.074 mm; F2—fraction >0.074 mm and <1 mm; min—minimum; max—maximum; med—median; SD—standard deviation; nd—not detected.

**Table 2 ijerph-17-01563-t002:** Estimated human health risk for elements As and Pb.

		F1					F2			
ID		ing	inh	drm	Total		ing	inh	drm	Total
**S1**	**As**	1.38 × 10^−4^	1.2038 × 10^−7^	2.0238 × 10^−5^	1.5838 × 10^−4^		1.3838 × 10^−5^	1.2038 × 10^−8^	2.0238 × 10^−6^	1.5838 × 10^−5^
	**Pb**	5.8538 × 10^−7^	1.5038 × 10^−10^	–	5.8638 × 10^−7^		2.6738 × 10^−7^	6.8738 × 10^−11^	–	2.6738 × 10^−7^
	**total**	1.3838 × 10^−4^	1.2038 × 10^−7^	2.0238 × 10^−5^	1.5938 × 10^−4^		1.4038 × 10^−5^	1.2038 × 10^−8^	2.0238 × 10^−6^	1.6138 × 10^−5^
**S2**	**As**	2.6838 × 10^−5^	2.3338 × 10^−8^	3.9338 × 10^−6^	3.0738 × 10^−5^		1.6138 × 10^−5^	1.4038 × 10^−8^	2.3638 × 10^−6^	1.8438 × 10^−5^
	**Pb**	6.2238 × 10^−7^	1.6038 × 10^−10^	–	6.2238 × 10^−7^		2.8938 × 10^−7^	7.4338 × 10^−11^	–	2.8938 × 10^−7^
	**total**	2.7438 × 10^−5^	2.3538 × 10^−8^	3.9338 × 10^−6^	3.1438 × 10^−5^		1.6438 × 10^−5^	1.4138 × 10^−8^	2.3638 × 10^−6^	1.8738 × 10^−5^
**S3**	**As**	2.1438 × 10^−5^	1.8638 × 10^−8^	3.1438 × 10^−6^	2.4638 × 10^−5^		9.1838 × 10^−6^	7.9938 × 10^−9^	1.3538 × 10^−6^	1.0538 × 10^−5^
	**Pb**	2.2438 × 10^−5^	5.76 × 10^−10^	-	2.24 × 10^−5^		5.71 × 10^−7^	1.47 × 10^−10^	–	5.71 × 10^−7^
	**total**	2.3738 × 10^−5^	1.92 × 10^−8^	3.14 × 10^−6^	2.68 × 10^−5^		9.75 × 10^−6^	8.13 × 10^−9^	1.35 × 10^−6^	1.11 × 10^−5^

F1—fraction <0.074 mm; F2—fraction >0.074 mm and <1 mm; ing—ingestion; inh—inhalation; drm—dermal.

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
