# Peer review of "Geochemical, Mineralogical and Morphological Characterisation of Road Dust and Associated Health Risks"

_ijerph, 2020, doi:10.3390/ijerph17051563_

Round 1

Reviewer 1 Report

A very well written paper with excellent English usage.

Only a few comments/questions:

line 187  Why such a high value of 1082 mg veh-1 km-1 for cobbled pavement?

For data in Table 1, please give limits of detection for the XRF method.  Also, please consider the number of sig figs for the respective element concentrations, although the SD helps to bracket these numbers.  Can you really give values for Al, for example, to 5 places?

Very small point, line 233, to read "three-way catalysts"

For Table 2, suggest you title the table "Calculated" or "Estimated" Human health risks for elements As and Pb. Given the factors used in the calculation with their respective ranges of uncertainty, and the data themselves, I think it is important not to present these numbers as certain values, but rather as calculated/estimated values.

Author Response

Please find attached our responses to the questions raised by the reviewer.

Reviewer 2 Report

A:  Road dust resuspension is one of the most serious way causing environmental pollution in the modern society. And it also has been one of the main air quality management challenges in the world. Therefore, this may be a valuable research attempt. But there are still some problems needed to be modified, and has not yet been met the requirements for publication.

B:  Comments on the manuscript

The manuscript did not specify the exact sample quantity? It is important for the paper. If only 3 samples (Inferred from the article content), it is too little. With so few samples, readers are not convinced that the conclusion of this paper is reliable.

It is necessary to add a figure to describe the spatial distribution of road dust sample in the manuscript.

Please improve the figure 1, figure 2 and figure 4 in the manuscript according to the guide for authors.

Please add the label number after each formula that appears in this manuscript.

Author Response

(The authors gave the same response as above.)

Reviewer 3 Report

General comments:

This MS investigated the load and particle size, chemical composition, minerals, and health risks of street dust that collected at three sites in a city of Portugal, the results is much helpful to trace the source of different elements in street dust of this city and conducive to understand the health risk level of the local habitants. The discussion was sufficient and convincible, and a minor revision is recommended.

Detailed comments:

In section 2.1, it’s need to clarify how many samples (number) was collected for each site, and how much (gram) for one sample got? In section 2.3 and 2.4, please give the detailed information of some important parameters, like the Earth’s crust individual elemental composition, RfDoute, CSF, ABSgi, IUR of each elements that involved in the calculations, in the supporting information part for easier understanding; For Table 1, it’s better to use unit % for the major elements, and show the detailed elemental composition of these three sites in the supporting information part, since the authors compared the content of some elements among these three sites but I could not differentiate this from Table 1; In section 3.4, it’s better to give explanation for different panels in Figure 2, either in the figure caption or in the context; In section 3.6, please give the detailed information (in Tables) of HQ that composed of different exposure routes for each element and HI, and both for children and adults, in the supporting information, for that higher than 1, please showed in bold. It’s impossible to judge which exposure route is more important,and for which cohort, children or adults, from Figure 4; And for the Tables and Figures showed in the supporting information, please cite in the text.

Author Response

(The authors gave the same response as above.)

Round 2

Reviewer 2 Report

Although the revised manuscript has been improved, it still does not meet the publication requirements of ijerph. And the high fees for samples collection can not be the reason to reduce the scientific conclusion of the manuscript.

Author Response

See attached file, please.
